# Personalised Dosing Using the CURATE.AI Algorithm: Protocol for a Feasibility Study in Patients with Hypertension and Type II Diabetes Mellitus

**DOI:** 10.3390/ijerph19158979

**Published:** 2022-07-23

**Authors:** Amartya Mukhopadhyay, Jennifer Sumner, Lieng Hsi Ling, Raphael Hao Chong Quek, Andre Teck Huat Tan, Gim Gee Teng, Santhosh Kumar Seetharaman, Satya Pavan Kumar Gollamudi, Dean Ho, Mehul Motani

**Affiliations:** 1Yong Loo Lin School of Medicine, National University of Singapore, Singapore 117597, Singapore; amartya_mukhopadhyay@nuhs.edu.sg (A.M.); lieng_hsi_ling1@nuhs.edu.sg (L.H.L.); gim_gee_teng@nuhs.edu.sg (G.G.T.); 2Medical Affairs—Research Innovation & Enterprise, Alexandra Hospital, National University Health System, Singapore 159964, Singapore; 3Division of Respiratory and Critical Care Medicine, Department of Medicine, National University Hospital, Singapore 119074, Singapore; 4Department of Health Sciences, University of York, York YO10 5DQ, UK; 5Department of Cardiology, National University Heart Centre, Singapore 119074, Singapore; 6Department of Electrical & Computer Engineering, National University of Singapore, Singapore 117583, Singapore; e0003601@u.nus.edu (R.H.C.Q.); motani@nus.edu.sg (M.M.); 7Division of Endocrinology, Department of Medicine, National University Hospital, Singapore 119074, Singapore; andre_th_tan@nuhs.edu.sg; 8Chronic Programme, Alexandra Hospital, National University Health System, Singapore 159964, Singapore; 9Division of Rheumatology, Department of Medicine, National University Health System, Singapore 119074, Singapore; 10Healthy Ageing Programme, Alexandra Hospital, National University Health System, Singapore 159964, Singapore; mdcsnks@nus.edu.sg; 11Division of Geriatric Medicine, Department of Medicine, National University Hospital, Singapore 119074, Singapore; 12FAST Programme, Alexandra Hospital, National University Health System, Singapore 159964, Singapore; satya_pk_gollamudi@nuhs.edu.sg; 13Division of Advanced Internal Medicine, Department of Medicine, National University Hospital, Singapore 119074, Singapore; 14Department of Biomedical Engineering, National University of Singapore, Singapore 119077, Singapore; biedh@nus.edu.sg

**Keywords:** chronic disease management, ambulatory care, self-management, artificial intelligence, personalised medicine

## Abstract

Chronic diseases typically require long-term management through healthy lifestyle practices and pharmacological intervention. Although efficacious treatments exist, disease control is often sub-optimal leading to chronic disease-related sequela. Poor disease control can partially be explained by the ‘one size fits all’ pharmacological approach. Precision medicine aims to tailor treatments to the individual. CURATE.AI is a dosing optimisation platform that considers individual factors to improve the precision of drug therapies. CURATE.AI has been validated in other therapeutic areas, such as cancer, but has yet to be applied in chronic disease care. We will evaluate the CURATE.AI system through a single-arm feasibility study (*n* = 20 hypertensives and *n* = 20 type II diabetics). Dosing decisions will be based on CURATE.AI recommendations. We will prospectively collect clinical and qualitative data and report on the clinical effect, implementation challenges, and acceptability of using CURATE.AI. In addition, we will explore how to enhance the algorithm further using retrospective patient data. For example, the inclusion of other variables, the simultaneous optimisation of multiple drugs, and the incorporation of other artificial intelligence algorithms. Overall, this project aims to understand the feasibility of using CURATE.AI in clinical practice. Barriers and enablers to CURATE.AI will be identified to inform the system’s future development.

## 1. Introduction

Chronic disease management typically requires long-term adherence to medication and lifestyle modifications [1]. While substantial efforts have gone into finding effective drug therapies and developing self-management support interventions, many chronic disease patients remain sub-optimally controlled [2,3]. One of the most significant limitations of today’s treatment regimens is the ‘one size fits all’ approach to disease management. For instance, drug dosing regimens follow standardised guidelines derived from population-based studies. These studies often do not include the diverse profile of patients clinicians face daily and thus the derived clinical guidelines are not fully representative [4]. While clinicians may use some discretion when treating patients, drug dosing decisions continue to be based on a limited number of variables, such as age, major organ function and severity of the disease. However, treatment success is dependent on a much broader spectrum of factors; genetics, environmental conditions, patient characteristics, and pharmacology can all influence an individual’s response to therapy [5,6,7]. Taking account of a greater degree of individual-level factors, tailored therapies or ‘precision medicine’ has emerged as an approach to improve disease management. 

Although precision medicine is generally regarded as a modern concept, the first known example can be traced back to blood group identification in 1901, leading to successful blood transfusion practices [8]. More recently, many advancements in the precision medicine field have been attributed to the human genome project. Mapping the human genome has substantially advanced knowledge on the role of genes in health and disease [9]. By accounting for individual variability, precision medicine can improve treatment efficacy and reduce unnecessary tests or adverse reactions [10]. However, there is still a pressing need to develop approaches and technologies that consider the spectrum of factors unique to individuals. Artificial intelligence (AI) may be one route to utilise a greater breadth and complexity of data to optimise treatment decisions. 

Amongst the many unique features of AI, a unifying concept revolves around the fact that AI can reconcile large amounts of data into actionable care management strategies. Applications in digital pathology and robotic surgery have been widely publicised [11]. In chronic disease, examples of AI applications include digital health programmes [11], conversational agents (e.g., chatbots) [12], games to promote physical activity [13], clinical decision support systems [14], wearables (i.e., to track disease management, physical activity, or disease exacerbations) [15], diagnostics, and prediction of chronic disease complications [16]. A novel dosing optimisation system—CURATE.AI—is the latest innovation that has the potential to improve chronic disease care. 

CURATE.AI is an actionable dosing optimisation platform, initially developed to improve the precision of chemotherapy dosing. The algorithm considers the treatment response over time, predicting the dosing needs dynamically to maintain the required levels of efficacy and safety. CURATE.AI has been validated in other disease indications [17,18,19] (e.g., oncology, immunosuppression, and infectious diseases) but has yet to be applied in patients with chronic disease. This study aims to demonstrate the feasibility of using an artificial intelligence-guided system to optimise medication dosing in hypertensive and type II diabetic patients. The study objectives are:To assess the clinical outcomes of patients treated using CURATE.AI technology.To evaluate the staff experiences of implementing and using the CURATE.AI system in clinical practice and identify the facilitators and barriers to its use.To elicit the opinions and experiences of patients receiving CURATE.AI guided care.To optimise the CURATE.AI algorithm through exploratory analysis, including artificial intelligence algorithms.

## 2. Materials and Methods

We will evaluate the CURATE.AI system in clinical practice through a single-arm feasibility study. The evaluation will be guided by modified versions of ‘The Non-adoption, Abandonment, Scale-up, Spread, and Sustainability’ (NASSS) and the Higgins & Madai (2020) frameworks on the development of artificial intelligence systems in healthcare. Quantitative and qualitative data will be collected through: (i) prospective collection and analysis of clinical data and (ii) staff and patient interviews. Exploratory analysis will investigate how best to optimise the algorithm using retrospective patient data.

### 2.1. The CURATE.AI System 

Details of the CURATE.AI algorithm have been published before [17,18,19,20,21]. In brief, CURATE.AI represents the drug input and output relationship by a second-order polynomial equation. The algorithm is calibrated to each patient using three input–output data points, creating a parabola. The parabola determines the drug inputs required to keep a patient within the output range. The parabola is continuously calibrated as further data are entered (Equation (1)). 

In this study, the input refers to the drug dose, and the output corresponds to blood pressure, HbA1c, or blood glucose. The algorithm can optimise multiple drugs simultaneously, although initially, we will only optimise a single principal drug. 

Equation (1) CURATE.AI algorithm.
(1)$$R\left(C,t\right)=F\left(S^{\prime },\;C,\;t\right)-F\left(S^{\prime },\;t\right)=x_{0}+\sum x_{i}c_{i}+\sum y_{ii}c_{i}^{2}+\sum z_{ij}c_{i}c_{j}\;$$

*R* (*C*, *t*) represents the overall treatment response; *F* (*S′*, *C*, *t*) represents a diseased patient under treatment; *F* (*S′*, *t*) represents the diseased patient; *S′* comprises the disease mechanisms; *C* represents the drug type and dose. Finally, *t* shows that every term in the series can vary with time and should be continually re-calibrated with clinical or point-of-care data. With regard to CURATE.AI-driven patient calibration, *x_i_* is the patient response coefficient to drug *i* at concentration *c_i_*, *y_ii_* represents the second-order response to the drug concentration *c_i_*, and *z_ij_* is the patient response coefficient to the interaction of drug *i* and drug *j* at their respective concentrations. To implement CURATE.AI, the values of *x*_0_, *x_i_*, *y_ii_*, and *z_ij_* are clinically/experimentally determined by calibrating phenotypic outputs of a specific patient and the drug dose inputs, leading to a personalised drug–dose combination for a specific patient.

### 2.2. Retrospective Study

CURATE.AI has been previously validated in retrospective and prospective studies [20] but has not been used in chronic disease states. We will conduct a retrospective analysis of patient data for two purposes. Firstly, an optimised dosing schedule will be generated using historical patient data on dose–response relationships. The aim is to demonstrate that a dosing schema can practically be generated and is clinically acceptable. 

Secondly, the dataset will be used to explore different ways of optimising the CURATE.AI algorithm. Examples include the inclusion of other covariates, which may impact the input–output response; the optimisation of multiple drugs simultaneously; whether calibration can be achieved with fewer calibration data points; and the use of different statistical models to forecast output responses, such as linear or polynomial regression or neural network regression.

For the exploratory analyses, outpatient records (hypertensives and type II diabetics) from 1 June 2018 to 30 September 2020 will be retrospectively extracted from Alexandra Hospital and the National University Hospital. Both institutions contribute to the same Electronic Health Record (EHR), which captures the patients’ health journeys across different health care providers. Details of medical histories, inpatient admissions, outpatient clinic visits, diagnostics, and medication prescriptions are captured by the EHR.

Factors associated with treatment response in patients with hypertension and type II diabetes are well reported [22,23,24,25,26]. Where possible we will extract these data. The following variables will be extracted: age, gender, race, marital status, residency status, postal code, diagnoses, subsidy status, visit date, referring institution, procedures or investigations and corresponding results, Charlson comorbidity index, and medication.

### 2.3. Prospective Study

#### 2.3.1. Participants

Forty patients (*n* = 20 hypertensives and *n* = 20 type II diabetics) will be recruited from the outpatient clinic at Alexandra Hospital, Singapore, during their routine clinic visits. The primary physician will identify potentially eligible participants and notify the research team. A research team member will discuss the project with the participant, screen for eligibility, and take consent. If a patient has both type II diabetes and hypertension, the treating physician will decide which study arm the patient should enrol on. Patients may be newly diagnosed or be those with established but poorly controlled disease. The presence of comorbidities and all prescribed medications will be tracked. The eligibility criteria are as follows: 

##### Inclusion Criteria

Adult patients (≥21 years) with clinically diagnosed type II diabetes or hypertension.

Expected to be followed up at Alexandra Hospital in the next six months.

##### Exclusion Criteria

Patients with cognitive impairment.

Patients with active cancer undergoing chemotherapy.

Patients on haemodialysis or peritoneal dialysis (which can cause rapid fluctuation of BP and blood sugar, respectively).

Pregnant patients.

Patients with type II diabetes and hypertension whose medications are changed simultaneously during their first clinic visit.

Patients with controlled HbA1c or blood pressure at baseline.

Serious concomitant disorders that would compromise the safety of the patient or their ability to complete the study. This may include a recent occurrence of stroke or sub-arachnoid haemorrhage in hypertensive patients, which necessitates tighter control, and occurrence of renal failure or lactic acidosis in patients with type II diabetes, which may warrant avoidance or reduction in metformin dose.

#### 2.3.2. Outcome Measures

We will report on clinical and qualitative outcomes relating to CURATE.AI’s effect, implementation, and acceptability. The following clinical outcomes will be reported:Proportion achieving blood pressure and glycaemic control at four months.Average number of days until blood pressure or glycaemic control is first achieved.Proportion relapsing after disease control is achieved (i.e., high blood pressure).Proportion compliant with clinic follow-up.Proportion compliant with home monitoring of blood pressure or glucose.Number of dropouts.Proportion of dosing decisions recommended by CURATE.AI but not implemented at the physician’s discretion and the magnitude of difference.

The study’s main purpose is to explore the feasibility of the CURATE.AI system in clinical practice and not to establish efficacy or effectiveness. The clinical outcomes are included to show the preliminary effects of the intervention. Qualitative work will seek to understand the practicalities of using the system in clinical practice, identify implementation barriers and enablers that can inform future development of the system, and to advance the processes associated with using CURATE.AI.

#### 2.3.3. Study Procedures

The patient flow is shown in Figure 1, and the study procedures with time points are detailed in Table 1. 

To optimise a single drug dosage, the CURATE.AI system requires three data points (drug/dosage input and corresponding response) to calibrate to each individual patient before dosing recommendations can be given. At the point of recruitment, the treating physician will identify a principal medication (that will be modulated by CURATE.AI), and the patient medical records will be mined for previous data on the principal drug use and corresponding treatment response. If previous data are unavailable (none or only partially available), clinic visits will be scheduled until three such data points are obtained. Once three data points have been collected, the patient will enter the study at baseline and will be followed up for up to four clinic visits. Data will be collected and entered into the algorithm by a research assistant (Table 1). The intention is to generate dosing recommendations live (i.e., within a clinic visit). Workflows will be assessed during the qualitative interviews with staff. 

The frequency of data collection (i.e., monthly or for home monitoring, two weeks after dose change) was established by mutual agreement by the study team based on two main points. Firstly, the clinical team indicated that a monthly follow-up frequency was the maximum that would be acceptable to patients without being burdensome. Secondly, although the drugs used to control hypertension and type II diabetes can exert their effects in a matter of hours, medications take time to reach a steady state and produce their maximum effect. For example, the European Society of Cardiology recommends increasing the dose for angiotensin converting enzyme inhibitors (a common drug used in hypertension) every two weeks when titrating [27], while weekly dose increases have been suggested for metformin use in type II diabetics [28]. The viability of the proposed data collection frequency will form part of the feasibility evaluation.

Socio-demographics (age, sex, race, marital status), lifestyle factors (diet and physical activity), medical history and other clinical variables will be collected at baseline and select clinic visits (Table 1). Medication history and changes in medication dose (other than the principal medication) will be collected and monitored throughout the study. Dietary practices will be assessed using the Dietary Practice Questionnaire (26 items). The questionnaire was previously used in Singapore’s National Population Health Survey [29]. Physical activity will be assessed using the Global Physical Activity Questionnaire (GPAQ), originally developed by the World Health Organization (WHO). The GPAQ has been validated in Singapore [30,31] and includes three domains: activity at work, travelling to and from places, and recreational activities. Current medication adherence will be assessed using the 9 item Hill-bone medication adherence scale, validated for use in chronic disease patients [32]. Renal function will be measured using the estimated glomerular filtration rate and liver function by measuring aspartate aminotransferase and alanine aminotransferase. Any disagreement between the CURATE.AI recommended dosing decision and the physician’s opinion will be documented. In cases of dispute, we will collect information on the final dosing decision and the reasoning behind the decision. At any point when the physician disagrees with the CURATE.AI recommendation, the physician’s opinion will supersede the CURATE.AI dosing decision. If a physician overrides a CURATE.AI decision, the participant will remain on the study until completion regardless.

Participants will be followed up monthly for four months. Clinically, blood pressure may be controlled with medication within four to six weeks [33]. For type II diabetes, although traditionally changes in HbA1c are assessed every three months, emergent evidence suggests clinically relevant changes may occur as early as one month [34]. Based on these data, four months was deemed appropriate to observe clinical improvements as well as being a sufficient period to test the system. Clinic visits may be face-to-face or virtual as appropriate. If a dosage change is required at a clinic visit, hypertensive participants will be asked to monitor their ambulatory blood pressure for a single six-hour period at home (once hourly). A six-hour ambulatory blood pressure recording is equivalent to a twenty-four-hour reading [35]. Patients will then continue to monitor blood pressure twice daily, morning and evening, for up to seven days. Following any change of type II diabetic medications, patients will be asked to wear a continuous glucose monitoring device for a minimum of seven days at home (the maximum wear time is fourteen days). Monitoring will commence fifteen days after the dose change is initiated (for either hypertensives or type II diabetics), so a steady state can be reached before assessment. These data will feed back into the CURATE.AI system, and further recommendations on dose adjustment will be generated if required. The treating physician will regularly review remote monitoring data for safety. If the patient’s medication is changed entirely, recalibration with CURATE.AI is required. At the last clinic visit, patients will be interviewed using a semi-structured survey on their experiences with the CURATE.AI system. 

#### 2.3.4. Implementation Evaluation 

The implementation evaluation will be guided by modified versions of ‘The Non-adoption, Abandonment, Scale-up, Spread, and Sustainability’ (NASSS) framework and the Higgins & Madai (2020) framework on the development of artificial intelligence systems in healthcare [36,37]. The NASSS framework consists of six domains: the condition, the technology, the value proposition, the adopter system, the health or care organisation(s), and the wider context. The AI framework comprises of four domains: clinical validation, regulatory affairs, data strategy, and model development. Each domain identifies the risks, objectives, types of results and key advice from product development to market launch. These frameworks will assist in identifying factors that can influence the implementation of the proposed CURATE.AI system and its effectiveness. As this is a feasibility study, we will only focus on domains relevant to this stage of development (Table 2).

##### Patient Survey

All study participants will be invited to complete an experience survey based on an adapted version of the Singapore outpatient experience survey. This survey is routinely administered at specialist outpatient clinics and covers topics on the clarity of communication, professionalism, and service organisation [38]. We have modified this instrument to include open-ended questions on the experiences of CURATE.AI-guided dosing, appointment attendance, and ease of home monitoring (Appendix A). If the participants cannot complete the survey or attend the last clinic appointment, they will be followed up with on the phone. A researcher will administer the interviews in English, Chinese, or Malay, according to patient preference. The survey will be piloted, for clarity, with volunteers not involved in the study before use.

##### Staff Interviews

Staff with experience of using the CURATE.AI system with at least one patient will be invited to participate in an in-depth interview. Participants will be approached in person or via email, emphasising that the interview is voluntary and that they may withdraw at any time. A purposive sample of participants will be recruited for maximum variance in professional roles and experience. Recruitment will occur on a rolling basis until data saturation has been reached. Interviews will be conducted using a semi-structured interview guide, including topics on the unmet needs of physicians when titrating medications in chronic disease patients, experiences of using the CURATE.AI system, perceived barriers and facilitators of using CURATE.AI in clinical practice, and how CURATE.AI could be integrated into existing workflows. Example questions include:Has CURATE.AI impacted your workload?What challenges have you faced when using CURATE.AI?What have been the expected and unexpected outcomes, both positive and negative, of using CURATE.AI?

The interviews will be conducted in a quiet, private meeting room, away from the clinic and ward areas, to encourage relaxed and open conversations. Interviews will be audio recorded and later transcribed for analysis.

#### 2.3.5. Data Analysis

All analyses will be conducted using STATA version 15.0 (STATA Corp, College Station, TX, USA). As appropriate, summary statistics will be presented as means with standard deviation, medians with interquartile ranges, and percentages. Clinical outcomes will be analysed separately for the hypertensive and type II diabetic groups. Blood pressure control is defined as a home or ambulatory blood pressure of less than 135/85 mmHg or office readings below 140/90 mmHg, unless otherwise defined by the treating physician [39]. Glycaemic control is defined as HbA1c < 7% [40] or glycaemic variability ≤ 36% or time in normal range (4.0 to 10.0 mmol/L) > 70% [41]. The proportion of patients achieving blood pressure control and glycemic control will be calculated by comparing the baseline reading to the four-month follow-up. A within-subject analysis of hypertensive and glycemic control will be conducted using the Chi^2^ or McNemar’s test, if cell counts are small (<5). The proportion achieving clinically relevant reductions in systolic and diastolic blood pressures by the four-month follow-up will be determined. The number of days until blood pressure or glycaemic control is first achieved will be calculated. The occurrence of relapse after control has been achieved (i.e., high blood pressure) will also be determined. 

Compliance with clinic follow-up will be calculated based on the number of scheduled sessions compared to attendance. Compliance with home monitoring will also be determined by remote receipt of data at scheduled time points. The proportion compliant with the home-monitoring schedule and the proportion of patients who drop out from the study will be reported. The number of disagreements between the CURATE.AI dosing recommendations and the physician’s opinion will be quantified. Success of the alignment on dosing decisions is defined as agreement with ≥70% of dosing decisions generated by CURATE.AI. The proportion of participants reaching this criterion will be derived. We will also summarise the reasons for disagreement and the magnitude of differences in dosing decisions, if it occurs. A sub-analysis will be conducted for participants whose dosing decisions were all based on CURATE.AI recommendations versus those who had one or more dosing decisions overridden by the treating physician. The purpose is to explore whether adherence to CURATE.AI confers an advantage in achieving clinical outcomes. Significance will be set at *p* < 0.05.

For interviews, audio recordings of the interview will be transcribed and translated into English (if necessary). A second independent researcher will check the accuracy of the translation. Data will be analysed using a thematic analysis method. Data will be coded according to the meaning of the sentences, followed by developing sub-themes and main themes. Qualitative data will also be organised according to the framework domains (Table 2).

## 3. Discussion

We aim to establish if the CURATE.AI system can practically be used for chronic disease patients and how it can be best operationalised through this feasibility study. We will explore if users find the system acceptable, identify factors that can help or hinder its use, and understand its potential for scaling. In addition, we will investigate ways to improve the algorithm. For example, the inclusion of other variables (i.e., patient characteristics), the simultaneous optimisation of multiple drugs, and the incorporation of other AI algorithms. The data generated from this study will inform the development of the CURATE.AI system and a future efficacy evaluation. Through the development of this application, we hope it may be possible to optimise dosing more rapidly, prevent adverse reactions, and reduce the burden of therapy.

### Potential Strengths and Limitations of the Study

We have chosen a mixed-method approach to explore the practicalities of using a precision drug dosing algorithm in clinical practice. Including qualitative methods, in addition to clinical outcomes, will bring a greater breadth and depth of understanding as to the use of CURATE.AI in clinical practice. However, there are limitations. This is only a feasibility study, and there is no control group. We will not establish the efficacy of the intervention from this study alone. Future randomised controlled trials will be needed to understand the clinical impact of CURATE.AI. Furthermore, the current iteration of CURATE.AI is not integrated within the existing electronic medical record systems. If CURATE.AI were to scale, workflows need to be developed so that data can seamlessly enter the algorithm. Qualitative interviews will explore how CURATE.AI could be adopted into existing workflows and systems. 

## 4. Conclusions

In conclusion, this study is the first in a programme of research that will seek to understand if CURATE.AI can effectively be used to manage chronic disease patients in the clinical setting.

## Figures and Tables

**Figure 1 ijerph-19-08979-f001:**
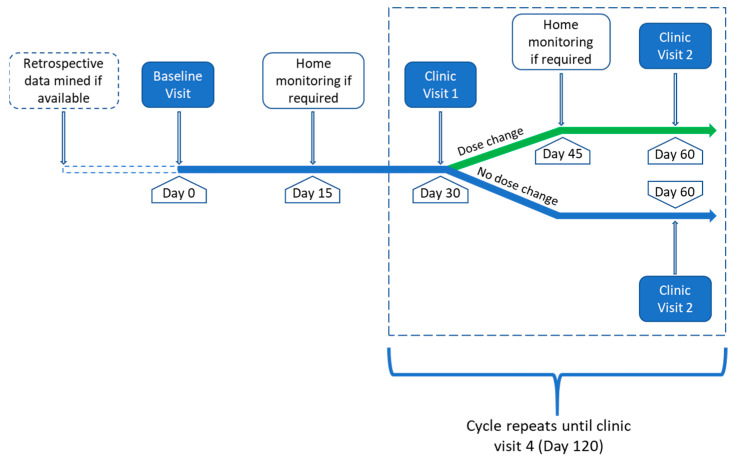
Patient flow.

**Table 1 ijerph-19-08979-t001:** Study procedures and timepoints.

Data Variables	Baseline ^a^	Visit 1 ^a^	Visit 2 ^a^	Visit 3 ^a^	Visit 4
**CURATE.AI variables:**					
Retrospective dose-response data mined	X				
Hba1C ^a^ or blood glucose	X	X	X	X	X
Blood pressure	X	X	X	X	X
**Other data captured:**					
Demographics	X				
Weight, BMI, hip-waist ratio	X	X	X	X	X
Medical history (morbidities, medication)	X				
Renal function	X				X
Liver function	X				X
Details of physical activity (16 items)	X				X
Details of diet (26 items)	X				X
Medication adherence (9 items)	X				X
Change in medications	X	X	X	X	X
Disagreements between CURATE.AI and physician	X	X	X	X	X
Patient survey					X

^a^ Home-monitoring may be scheduled 15 days after dose change, if indicated.

**Table 2 ijerph-19-08979-t002:** Modified NASSS and AI framework domains and associated data sources [36,37].

NASSS Domain	Data Sources
1A/1B: What is the nature of the condition?/What are the relevant sociocultural factors and comorbidities?	Patient profiles and patient interviews
2A: What are the key features of the technology?	The algorithm and desired features identified in staff interviews
2B: What kind of knowledge does the technology bring into play?	Application of algorithm in chronic disease care, and staff and patient interviews
2C: What knowledge and support are required to use the technology?	Patient and staff interviews
3B: What is the technology’s desirability, efficacy, safety, and cost effectiveness?	Study outcomes, and patient and staff interviews
4A: What changes in staff roles, practices, and identities are implied?	Staff interviews
**AI Framework Domain**	**Data Sources**
Data form: Data access, the structure of data, appropriateness of data, data plan	Evaluation of data collection from medical record data, clinic assessments, and home-monitoring
Model development form: Determine the best algorithm type, scalability of the algorithm	Algorithm development through retrospective analysis of medical record data
Model development build: Pilot test performance of the algorithm	Study outcomes and staff interviews

## Data Availability

Not applicable.

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
