# Peer review of "Personalised Dosing Using the CURATE.AI Algorithm: Protocol for a Feasibility Study in Patients with Hypertension and Type II Diabetes Mellitus"

_ijerph, 2022, doi:10.3390/ijerph19158979_

Round 1
Reviewer 1 Report
This protocol describes the plan to refine, implement and evaluate an AI-based dosing platform for patients with chronic disease, diabetes and hypertension. These platforms have the potential to contribute to delivering personalized medicine for people with chronic diseases in order to optimally manage their medication regimens and improve quality of life. I commend the authors for a very thoughtful evaluation using both quantitative and qualitative assessments. I have several questions about the integration of the system and the methodology used for evaluating.
- Problem statement is not developed enough: I would like to be presented with evidence that doctors in the pilot hospitals feel that they are not able to titrate/prescribe proper dosage effectively enough for their patients. And consequently that they think an AI-generated clinical support system would help them. AI systems often face the challenge of low physician adoption and motivation as well as few health care personnel supporting the integration. Before this is implemented it would be helpful to discuss that there has been due diligence with the users and support staff for the algorithm.
- Further, how will the curate.AI tool insights be generated for doctors? How will it be integrated into their workflow? Further, which data will be relayed into the algorithm system for real time feedback? And, is it technically feasible? Second, is it feasible from a workload standpoint or will it all be automated? Please clarify.
- This protocol involves a very large data intake – can it be discussed which data will need to be regularly collected to feed into the system once AI the system is up and running? Is it reasonable to think this can be incorporated into a doctors ongoing workload?
- Does the AI system include side-effect data in its modeling?
- It would seem to be important to include a control group that is not exposed to the AI tool to compare patient outcome trajectories over multiple time points for the evaluation. It would seem that a key outcome is whether participants can maintain control over an extended period of time. Can participants be controlled (A1c of BP) at baseline or is this an exclusion criteria?
- It would seem important that participants are stratified by disease duration in the evaluation as it is a major confounder of dosage and the outcomes, can you comment on this?
- The retrospective study will collected patient records (of diabetics and hypertensives) during 2019 from Alexandra Hospital 124 and the National University Hospital, Singapore outpatient clinics. What is the sample size of the relevant population over a year? Is it large enough to train test and validate a robust AI engine? Is the EHR data comprehensive and high quality enough for these analytics? Does it include data on history of medications and response as these are key baseline parameters for informing present meds and dosage.
Author Response
Thank you for the feedback. The points raised have helped us to improve the quality of the manuscript. We have addressed each point in the attached file.

Reviewer 2 Report
CURATE.AI has been the object of a prior publication and validated in several clinical areas, but not for chronic diseases. The authors propose a protocol to assess the feasibility of CURATE.AI for diabetes type-II and hypertension, which are two highly prevalent chronic diseases. It is a feasibility study rather than a randomized controlled trial, so the authors are clear that it is not intended to establish efficacy. Overall, the protocol is well-written, but numerous aspects should be made more precise. Each one of these aspects can be addressed quickly, but since there are several I've categorized this revision as 'major'.
1) The equation on line 99 (which should be numbered Equation 1) is too short of a description for the entire CURATE.AI system. A paper should be self-contained, so readers should not be forced to read the other paper to understand this one. At least one guiding example originating from diabetes or hypertension treatment would help, preferably relying on a visual similar to Figure 1 in ref 14.
2) Lines 108-109, several variables will be 'calibrated'. Please explain (i) the calibration process and the (ii) total number of variables that would have to be calibrated given n drugs. Also, you're only looking at 2nd order interactions between drugs; please comment on (iii) whether 3rd order interactions (or beyond) may occur in your application context.
3) The previous paper on CURATE.AI had about 120 references, so it was extensively based on the literature. This protocol only has 26 references, which are relevant but very light to cover the application area. The idea that AI and precision medicine have seen "limited advances in the chronic disease field" is a vast over-simplification that ignores at several active areas of research within the broader matter of computer-tailored interventions (see https://journals.sagepub.com/doi/full/10.1177/2055207618824727 for a sample scoping review focusing on one AI technology and https://www.healthpsychologybulletin.com/article/10.5334/hpb.26/ for the increasing use of AI in ecological momentary assessment for health). As additional examples of specific areas, first, games for health have often been used for eating behaviors and physical activity behaviors (both of which ultimately contribute to diabetes type-II and hypertension); the games are an indirect vehicle to elicit patients' perspectives and educate them (i.e., a health intervention). As examples, see https://www.jmir.org/2020/4/e14549/, https://journals.sagepub.com/doi/full/10.1177/1460458214521051, or the review in https://www.liebertpub.com/doi/full/10.1089/g4h.2018.0024. Second, simulations are routinely used to create tailored treatment plans instead of the problematic 'one-size-fits-all'; see https://bmcmedresmethodol.biomedcentral.com/articles/10.1186/1471-2288-14-130 for example, https://www.sciencedirect.com/science/article/pii/S1532046410000882, or https://www.sciencedirect.com/science/article/pii/S1532046418302119. Finally, machine learning have a growing use in precision medicine (e.g., https://onlinelibrary.wiley.com/doi/abs/10.1111/obr.12667 for obesity), particularly to predict when to switch drug, and to which cocktail; see https://ieeexplore.ieee.org/abstract/document/9552036/ as an example, as well as https://link.springer.com/article/10.1007/s11904-020-00490-6 for complementary facets of AI in managing the chronic condition of HIV. In fact, there is such a vast literature on the topic that one may get lost trying to process it all. A two paragraph summary with a few examples for each area (health games, simulation, machine learning) would suffice, and hence clarify that AI and precision medicine have really been around for a while in chronic disease management.
4) Line 115 suggests a 'theoretically optimized dosing schedule'. If I understand the technology correctly, it is reactive more than predictive. You'll know how a patient reacts to a drug (and to interactions) after administering them. You won't know ahead of them what drug a patient may need, and when. So it is not a theoretical optimum, but an experimental optimum.
5) Line 120 "exploration of other AI based approaches" is a very vague statement. Please clarify why you need to explore other approaches, and give examples of those under consideration.
6) Lines 125-127, what's the rationale for inclusion of some of these variables with regard to treatment outcomes? For example, why include postal code, residency status, subsidy (presumably health insurance?), referring institution? There should be a clear clinical relevance of the data captured.
7) Please clarify early on (title, abstract) that this is about type-II diabetes (the most common, acquired in part from lifestyle factors) rather than type-I.
8) Lines 154-155 are problematic in a protocol. It leaves it entirely to the investigator's discretion to decide whether the patient is at risk, which is an open-ended form of "just trust us". There should be greater clarity about the types of disorders that would unequivocally qualify, rather than leave it up to individual practitioners.
9) Line 162, please clarify the 'clinically relevant reductions' of interest here.
10) Line 169, it is a great idea to check when the physician made a different decision. I suggest that the authors collect (i) the decision made by the physician, and (ii) an open-ended rationale for making a different decision. This information would be essential to improve the system. If the authors decline, with the right arguments, it could then be included as limitations instead.
11) Lines 173-175 (and previous parts of the article) refer to the 'clinical acceptance of the algorithm'. But ultimately, it's only a measure of clinical agreement with the algorithm, not acceptance. Acceptance would include how well the tool integrate in the clinician's workflow. Please refrain from using 'acceptance'.
12) Lines 175-176, the system will be deemed successful if decisions are in agreement over 70% of the time. Why this criterion? If the other 30% of the time, the clinician has a vastly different decision than the system, then it's a much bigger problem than if the decision has minor differences. I would encourage the authors in (i) justifying their criterion and (ii) complementing it with a measure that assess the magnitude of the difference rather than just its frequency.
13) Lines 199-200, there is an argument for a monthly frequency in part because "medications take time to reach a steady state and produce their effects". Please provide evidence that the medications administered for type-II diabetes or hypertension would need one month to reach a steady state.
14) Table 1 is supposed to list the variables, but some items are individual variables whereas others are entire questionnaires. I suggest that when an entire questionnaire is listed, the authors include in parentheses the number of variables that it tracks.
15) Line 246, "participants will be followed up monthly for four months": please provide evidence that this is a sufficient duration in the context of these chronic diseases.
16) Lines 282-284, the authors modified the questionnaire. If you modified it, we need to (i) see what it looks like now, and have a clear explanation of (ii) what you modified as well as (iii) why it was modified.
17) Line 287, "Data collection tools will be piloted before use." How?
18) Lines 289-296. (i) Please clarify that you plan to get informed consent from staff. They are human subjects in your study so they cannot be forced to enroll or experience negative consequences if they do not provide certain answers. (ii) Could you please provide examples of the questions that would be posed during the semi-structured interview? These interviews tend to start with the same set of seed questions because being more customized to each interviewee. (iii) How many staff interviews will be conducted, and how do you determine this number as sufficient for your analysis?
19) There should be power calculations for the size of the sample used here (20 + 20).
Minor notes:
- missing closing quote for the one opened on line 88
- line 96 says that details of CURATE.AI have been 'published' elsewhere. 14 is a publication, but 15 would best be called a description rather than a publication (which implicitly suggests peer-reviewed).
Author Response
We are very grateful for the review and feedback. We have addressed the points raised in the attached file.

Round 2
Reviewer 1 Report
See attached - my comments in red.

Author Response
Thank you for the further comments. Please find attached our responses.

Reviewer 2 Report
The authors have provided a clear and detailed response letter. The primary weakness of insufficient explanations in the previous version has been adequately addressed. The review of the literature is rather succinct, but it is sufficient for a protocol. I now support the execution of this protocol and I look forward to a future manuscript detailing its results.
Author Response
Thank you again for the feedback on this manuscript.